# Tuberous Sclerosis Complex Axis Controls Renal Extracellular Vesicle Production and Protein Content

**DOI:** 10.3390/ijms21051729

**Published:** 2020-03-03

**Authors:** Fahad Zadjali, Prashant Kumar, Ying Yao, Daniel Johnson, Aristotelis Astrinidis, Peter Vogel, Kenneth W. Gross, John J. Bissler

**Affiliations:** 1Department of Clinical Biochemistry, College of Medicine & Health Sciences, Sultan Qaboos University, Muscat 123, Oman; fahadz@squ.edu.om; 2Department of Pediatrics, University of Tennessee Health Science Center and Le Bonheur Children’s Hospital, Memphis, TN 38103, USA; pkumar21@uthsc.edu (P.K.); yyao@uthsc.edu (Y.Y.); aastrein@uthsc.edu (A.A.); 3Children’s Foundation Research Institute (CFRI), Le Bonheur Children’s Hospital, Memphis, TN 38103, USA; 4Molecular Bioinformatics Center, University of Tennessee Health Science Center Memphis, TN 38103, USA; djohn116@uthsc.edu; 5Department of Veterinary Pathology, St. Jude Children’s Research Hospital, Memphis, TN 38105, USA; Peter.Vogel@StJude.org; 6Department of Molecular and Cellular Biology, Roswell Park Cancer Institute, Buffalo, NY 14263, USA; Kenneth.Gross@RoswellPark.org; 7Department of Pediatrics, St. Jude Children’s Research Hospital, Memphis, TN 38105, USA

**Keywords:** TSC complex, extracellular vesicles (EVs), mTORC1, renal cyst

## Abstract

The tuberous sclerosis complex (Tsc) proteins regulate the conserved mTORC1 growth regulation pathway. We identified that loss of the *Tsc2* gene in mouse inner medullary collecting duct (mIMCD) cells induced a greater than two-fold increase in extracellular vesicle (EV) production compared to the same cells having an intact *Tsc* axis. We optimized EV isolation using a well-established size exclusion chromatography method to produce high purity EVs. Electron microscopy confirmed the purity and spherical shape of EVs. Both tunable resistive pulse sensing (TRPS) and dynamic light scattering (DLS) demonstrated that the isolated EVs possessed a heterogenous size distribution. Approximately 90% of the EVs were in the 100–250 nm size range, while approximately 10% had a size greater than 250 nm. Western blot analysis using proteins isolated from the EVs revealed the cellular proteins Alix and TSG101, the transmembrane proteins CD63, CD81, and CD9, and the primary cilia Hedgehog signaling-related protein Arl13b. Proteomic analysis of EVs identified a significant difference between the *Tsc2*-intact and *Tsc2*-deleted cell that correlated well with the increased production. The EVs may be involved in tissue homeostasis and cause disease by overproduction and altered protein content. The EVs released by renal cyst epithelia in TSC complex may serve as a tool to discover the mechanism of TSC cystogenesis and in developing potential therapeutic strategies.

## 1. Introduction

More than one million patients world-wide suffer from tuberous sclerosis complex (TSC) and have mutations in either of the tumor suppressor genes, *TSC1* or *TSC2* [1]. Together, the TSC proteins regulate mTORC1 activity and control cell growth [2], a function critical for organogenesis and organ maintenance [3,4]. The detection of TSC renal disease is highly dependent on the imaging modality but all postmortem samples have renal disease [5]. Approximately 40% of patients with TSC experience a premature decline of glomerular filtration rate (GFR) in the absence of angiomyolipomata bleeding or interventions [6]. The loss of renal function is caused by the five different TSC renal cystic disease phenotypes, and angiomyolipomata [1]. 

Although there is ample scientific data associating cystogenesis to the mTORC1 pathway, results from animal model and human tissue studies are difficult to reconcile with the accepted dogma about TSC-associated renal disease pathogenesis, which is based on findings in angiomyolipomata. In this lesion, a somatic mutation (second hit) mechanism of TSC renal disease [7] results in an inactivating mutation and loss of heterozygosity with the loss of tuberin staining [8]. However, in mouse models studied there is an incongruity between elevated cystic epithelial mTORC1 activity (phospho-S6 expression) and the very low percentage of cells exhibiting loss of *Tsc* expression [9]. The accepted second hit mechanism is difficult to reconcile with murine *Tsc* cystic disease because these investigations fail to find somatic mutation in a majority of cysts, indicating that the majority of renal cysts maintain their *Tsc* locus integrity [9,10], and because human cysts also continue to express tuberin and hamartin [11]. Such a low rate of loss of heterozygosity is seen also in *PKD1* associated autosomal dominant polycystic kidney disease, suggesting that such cystic disease may represent a unique disease mechanism [12,13]. 

In a mouse model that exhibited a specifically targeted deletion of *Tsc2* in the principal cells by using the aquaporin-2 promoter to drive the expression of Cre-recombinase, we found that cysts were composed almost entirely of intercalated cells [14]. These intercalated cells maintained the *Tsc2* locus and exhibited increased mTORC1 activity [14]. The mechanism of phenotype spreading between Tsc2-null cells and intact cell is mediated by extracellular vesicles (EVs) [14]. This phenotype spreading model was also demonstrated in neuronal cells of tuberous sclerosis [15]. To understand how the principal cell induced the intercalated cells to form the cyst, we used a cell culture model where we used inner medullary collecting duct cells with or without a CRISPR/CAS9-mediated deletion of *Tsc2* to model the principal cells. We demonstrated that the loss of *Tsc2* gene function significantly increased the production of extracellular vesicles (EVs). These EVs have an altered proteomic profile that may increase recipient cell resistance to cellular stress and promote proliferation.

## 2. Results

### 2.1. Renal Collecting Duct Cells Produce EVs In Vitro

We isolated EVs from media of renal collecting duct cells (mIMCD) and *Tsc2*-deleted T2J cell lines using size exclusion chromatography (Figure 1). Using biophysical characterization techniques, including tunable resistive pulse sensing (TRPS) and dynamic light scattering (DLS), we characterized the EVs and we used transition electron microscopy (TEM) to demonstrate the EVs purity as well as morphology. We found size agreement of the EVs using all three methods (Figure 2a). Electron microscopy revealed high purity of the expected cup-shaped structures [16] of EVs in the isolates from both the mIMCD and the T2J cell lines (Figure 2a). The EVs isolated from mIMCD and T2J cell lines were heterogeneous in nature and had a mixed sized population of EVs. We were consistently able to isolate two-fold more EVs from the *Tsc2*-deleted T2J cell line compared to the parental mIMCD cell line (*p* < 0.05) (Figure 2b). Approximately 88%–91% of total EVs population were composed of small particles in size range of 100–250 nm. While less than 10% of the EVs population were greater than 250 nm as shown by TRPS (Figure 2c). The difference in particle distribution between mIMCD and T2J cells was not statistically significant.

To further verify that these structures were truly EVs, we used Western blot analysis to identify EV markers. The isolates from the conditioned media stained for EV markers such as the cellular proteins Alix and TSG101, transmembrane proteins such as CD63, CD81, and CD9, as well as the cilia protein ARL13b (Figure 2d). 

### 2.2. Loss of Tsc2 Increases the Production of EVs

To better understand how the *Tsc2* gene status increased the number of EVs, we examined their synthetic rate. To assess this, we transfected the cell lines with a CD63-green fluorescent protein (GFP) construct and used total protein to normalize the fluorescence. The transfection efficiency for both cell lines was equivalent (Figure 3a). To measure EV synthetic and release rates we used mean fluorescent intensity (MFI) normalized to the cell lysate protein concentration keeping time constant. Both the extracellular vesicle synthetic rate (Figure 3b, *p* < 0.05) and release rate (Figure 3c, *p* < 0.01) were increased in the *Tsc2*-disrupted T2J cell line, compared to parental mIMCD cells. 

### 2.3. Proteomic Analysis of EVs

Because mTORC1 activity regulates protein synthesis and we identified differences in the EV synthetic and release rates that were dependent on the *Tsc2* status, we posited that there could be *Tsc2*-dependent difference in the EVs proteins. To test this hypothesis, we performed proteomic analysis of the EVs originating from mIMCD and T2J cells. Because the cell lines were isogenic except for their *Tsc2* status, they had significant similarities. Biological processes enriched in mIMCD EVs include cellular proliferation, regulation of primary cilia, and cellular stress (Table 1). Both cell line EVs exhibited clusters and enrichment in proteins involved in metabolic pathways (Figure 4a,b). The biological process analysis also revealed proteins involved in proteolytic activity and cell division, include mini-chromosome maintenance proteins (Figure 4c). Additional pathways significantly increased in the mIMCD proteome included primary cilia and cellular response to stress associated proteins. We compared the proteins from these cell lines to the known EV protein databases. We compared 79 proteins to two public domain EV protein databases and found that the EVs from the mIMCD cell lines had 51 proteins that were not identified in the other databases (Figure 4d). The differences are more striking when our findings were compared to the top 100 proteins in the known databases (Figure 4e), where one protein was in common to all three data sets, and one protein was shared by one database, but the other 77 were not previously identified. 

There were also significant differences in the protein content between the cell lines. Specifically, there was significant suppression of four proteins in the *Tsc2* mutant cell line compared *Tsc2* competent mIMCD cell line (Table 2). These were all extracellular vesicle related proteins myosin-9 [17], T-complex protein 1 subunit γ [18], adseverin, and protein disulfide-isomerase A3 [19].

## 3. Discussion

The majority of TSC patient develop some form of kidney cyst during their life time but the exact mechanism of kidney cystogenesis in tuberous sclerosis complex is unknown [20]. The TSC cystic disease can manifest as one or more of at least five patterns of disease [1], and appears to have a very novel cystogenic mechanism involving the renal tubular cells [14]. In a previous study, we reported that the loss of *Tsc2* gene in a mouse renal principal cell affected the phenotype of intercalated cell causing an enhanced mTORC1 activity, thus leading to renal cystogenesis likely via an EV-dependent pathway [14] (Figure 5). To elucidate our hypothesis, we utilized a well characterized in vitro model.

In this current study, we isolated and characterized the EVs from isogenic cell lines differing only in the *Tsc2* gene status. We utilized well-established size exclusion chromatographic (SEC) techniques to isolate the EVs from the cell culture media (Figure 1) with a very high purity and free from protein contamination [21,22,23]. EVs were further characterized by various sources such as TRPS technology, DLS, and TEM. From all the characterization techniques we found the diameter of EVs ranged from 100 to 250 nm (Figure 2) [24]. The EVs demonstrated the expected shape and size by electron microscopy. Following the International Society for Extracellular Vesicles (ISEV) guidelines which suggests that characterization of EV protein content must include at least one cytosolic protein and one transmembrane protein [24], we identified the presence of the cellular proteins Alix, and interestingly TSg101 (Figure 2b). TSG101 is a more specific to small EVs [25] which was compatible with our DLS, TRPS, and TEM findings. Other tetraspanin proteins, such as CD63, CD81, and CD9 were also identified, indicating that the EVs biogenesis was through the multivesicular bodies (MVBs) [26]. We also identified the ciliary protein ARL13b. This protein is involved in hedgehog signaling [27,28], which in turn appears to be involved in cystogenesis [29] and even malignancy [30].

We demonstrated that the loss of *Tsc2* in a mouse inner medullary cell line drives a significant (almost two-fold) increase in the production of EVs (Figure 2b). To begin to understand this, we approached the problem using mass spectroscopy. Using independent isolations of mIMCD cells and the same cell with a disrupted *Tsc2* gene, we found that the loss of the Tsc2 gene significantly decreased four proteins. Interestingly, all four proteins have been identified in EVs in the past and are associated with vesicular trafficking. One explanation may be that these proteins are rate limiting and that the EV producing cells deplete them with the increase production rate in the *Tsc2* disrupted cell line. Another equally tenable possibility is that the significant reduction of the four proteins alters the EV binding, uptake, and trafficking in the recipient cell. The combined effects of altered EV production and uptake may drive resistance to cellular stress and proliferation and thus contribute to cystogenesis as well as malignancy [31].

We compared our 79 proteins to two public domain EV protein databases and found that the EVs from the mIMCD cell lines had 51 proteins that were not identified in the other databases (Figure 4d). The differences are more striking when our data were compared to the top 100 proteins (Figure 4e), where one protein was in common, and one protein was shared by only one database, but the other 77 were not previously identified.

We posit these differences between the EV protein databases and our EV proteins arise from the origin of the cell line we utilized. The mIMCD cell line was derived from a region in the kidney which has a remarkably low oxygen tension [32]. A large fraction of renal oxygen consumption is used to drive transport, including the reabsorption of more than 99% of the filtered sodium under normal physiologic conditions. Renal blood flow is highly compartmentalized, such that the cortical flow is substantially greater than that in the medulla. However, the medullary blood flow is further compartmentalized because the blood vessels that deliver oxygen to the medulla are sequestered within tightly packed vascular bundles, away from the renal tubules. This renal vascular architecture results in even lower oxygen tension outside the vascular bundles and contributes to hypoxic injury. This unique microenvironment may help explain some of the protein findings in the EVs, such as the enrichment in proteins involved in metabolic pathways and the adaption to cellular stress such as hypoxia. The presence of proteins involved in cell division, include mini-chromosome maintenance proteins also are not so surprising. Under normal conditions the kidney has a low cell turnover, but under conditions of injury, the renal tubule cell can robustly replicate in a repair process [33]. Given these results, it is less surprising that extracellular vesicles may be involved in ischemic preconditioning [34].

These results also are not entirely surprising given what is known about increased mTORC1 activity. The loss of *Tsc2* function in human as well as mouse cells induces significant endoplasmic reticulum stress [35,36]. Such stress is a potent inducer of EV production [37]. Adding to this stimulation, the increase in mTORC1 activity drives hypoxia inducible factor synthesis and release [38,39]. This hypoxia signaling also is a strong inducer of EV production [31,40].

These two features of *Tsc2* disrupted cells could facilitate the increased production of the EVs. A fundamental question raised by these findings is why such stress would increase the EVs that that we have identified. The protein functions enriched in the mIMCD EVs proteome included proteolytic activity and cell division (Table 1). Significant amounts of proteasome subunits are included and may be present to facilitate management of endoplasmic reticulum stress in the recipient cell, or possibly to facilitate converting proteins back into amino acids to enhance growth. The protein enrichment for cell division include mini-chromosome maintenance proteins. These are critical proteins for cell division and function as targets of various checkpoint pathways, such as the S-phase entry and arrest checkpoints. Deregulation of MCM function has been linked to genomic instability and a variety of carcinomas [41,42]. We posit that the data are compatible with a role of EVs that not only signal stress, but function in tissue maintenance such that stressed cells signal not only the stress, but also communicate survival strategies for the recipient cells, as well as signaling for the likely cellular replacement requirements (Figure 4). In this model, disease then is caused by dysregulated tissue repair rather than deregulated cellular proliferation. This would account for why cells with an intact TSC axis are involved in the disease phenotype [14].

While this manuscript focuses on the protein aspects of the mIMCD EVs and the effect of the loss of *Tsc2*, further work in this model is required to fully understand the differences induced by the loss of *Tsc2*. Ongoing experiments are focused on examining EV microRNA, long noncoding RNA, DNA, and even lipids. Our previous clinical study research found that rapamycin compounds could reduce the angiomyolipomata volume in TSC patients [43,44]. Interestingly, preclinical work suggests that a combination therapy of imatinib and rapamycin has more antitumor effect compared to rapamycin alone in an in vivo model of cutaneous tumorigenesis [45]. This improved effect could relate to the fact that tyrosine kinase inhibitors such as imatinib and dasatinib possess an inhibitory effect on EV production [46]. By understanding all the potential signaling avenues employed by EVs, we hope to better understand the fundamental pathobiology. This underpinning will be used in design better treatments and gain insight into related disease processes including malignancy.

## 4. Materials and Methods

### 4.1. Cell Lines

A mouse renal principal cell line derived from inner medullary collecting ducts (mIMCD) [47] (ATCC^®^ CRL-2123™) and its isogenic partner, T2J, which is identical except for a *Tsc2* gene disruption facilitated by CRISPR/CAS9 technology previously described [14]. The guide RNA was targeted at exon 4 of Tsc2 gene. The cell lines were maintained in complete growth media containing DMEM/F-12 (Invitrogen Cat 11330032) supplemented with 10% FBS and penicillin-streptomycin (Invitrogen) [14]. The culture was maintained at 5% CO_2_ atmosphere, humidified 95% air and 37 °C temperature.

### 4.2. Isolation of EVs by Size Exclusion Chromatography (SEC)

EVs were isolated from the culture media of mIMCD and T2J cells lines separately using a size exclusion chromatography column (Figure 1) and the manufacturer’s protocol [21,48]. Briefly, cells were maintained in DMEM/F-12 complete media up to 80% confluency. The cells were washed twice with PBS and incubated with serum-depleted media for 24 h. The conditioned cell culture media was collected and centrifuged at 2,000× *g* for 15 min at 4 °C to remove cell debris. The EV-containing supernatant was concentrated using a Millipore Amicon^®^ Ultra 15mL 10K centrifugal filter (Cat# UFC901008).

The SEC column and sample buffers were at room temperature. The column was flushed with one column volume of PBS. Then the concentrated cell culture medium was applied on the top of equilibrated qEV column (Izon Sciences) and fractions were collected. The EV rich fractions were eluted using particle free PBS. The EV containing fractions were pooled together and concentrated to ~500 µL using ultrafiltration. The EVs were analyzed using tunable resistive pulse sensing (TRPS), dynamic light scattering (DLS) and transition electron microscopy (TEM). EVs were stored at 4 °C for immediate use or stored at –80 °C for later use.

### 4.3. Characterizations of EVs

The EVs were analyzed using TRPS with the qNano gold instrument (Izon Science, Medford, MA). This approach provided a simultaneous result of size distribution and concentration. The instrument was calibrated using the polystyrene bead CPC100. The samples were allowed to pass through a nanopore NP150, which was stretched to 47 mm and a voltage of 0.34 mV. All samples were analyzed at 15 mbar pressure up to minimum 500 particles count for each, and each sample was analyzed in triplicate. Size distribution and concentration of EVs were obtained using Izon Control Suite v3.3 software. Data were collected and analyzed according to the manufacturer’s instruction.

EVs were further characterized for size using DLS and for size and purity using TEM. For DLS, 50 μL of EVs suspension were dissolved in 950 μL of ultra-pure water and analyzed using Zetasizer Nano-ZS (Malvern instrument, UK). To evaluate the isolated EVs morphologically for their size, shape and quality using TEM, we used negative staining. Briefly, 8 μL of fresh EVs were allowed to adsorb on to 200 mesh Formvar/Carbon coated grid for 20 min at room temperature followed by staining using 2% uranyl acetate. The excess stain was removed by washing the girds twice with PBS. The grids were visualized using JEOL 2000EXII TEM (Neuroscience Institute, The University of Tennessee Health Science Center).

### 4.4. Protein Isolation and Western Blot

To concentrate the EV protein for Western blot analysis, we pooled six elutions of EVs from each cell line. We precipitated the protein using polyethylene glycol 6000 (PEG) (Sigma Aldrich #81260) to a final concentration of 12% mixed with NaCl with a final concentration of 75mM. The EVs and PEG were mixed gently by inverting the tubes six times and incubated at 4 °C for 16 h, followed by centrifugation at 12,000× *g* for 45min at 4 °C in a tabletop centrifuge. The pellet was resuspended in PBS and used for SDS-PAGE.

The isolated proteins were resolved in 4%–20% Mini-PROTEAN^®^ TGX™ Precast Protein Gels from Bio-Rad Cat #456-1097 (Hercules, CA). Proteins were transferred to a polyvinylidene difluoride (PVDF) membrane by a semidry method at 15 V for 25 min. The membrane was blocked using 5% *w/v* BSA or 5% *w/v* nonfat dry milk, followed by overnight incubation with appropriate antibody at 4 °C. The following primary antibody is used: mouse anti-Alix (1:100, proteintech 12422-1-AP), mouse anti-TSG101 (1:1000, sc-7964), rabbit anti-CD63 (1:500, proteintech 25682-1-AP), rabbit anti-CD81 (1:1000, sc-18877), mouse anti-αCD9 (1:500, Thermo 14-0091-82), and rabbit anti-ARL13b (1:1000, ab83879). Goat anti-mouse or goat anti-rabbit conjugated with horseradish peroxidase were used as secondary antibodies and incubated at room temperature for one hour. The antibody-antigen complexes were detected using the chemiluminescence agent.

### 4.5. EV Synthesis and Release

In 12-well culture plates, equal numbers of mIMCD and T2J cells were transfected with eGFP-tagged CD63 (pcDNA3-EGFP, Addgene) using Lipofecatime plus reagent (Thermo Fisher Scientific). Transfection efficiency was measured by measurement of GFP signal to nuclear stained DAPI signal. Images were captured using Leica DMi8 fluorescence microscope. To measure eGFP-EV synthesis, cells were washed 48 h after transfections and then lysed in cell lysis buffer. Fluorescent intensity was measured at wavelengths 480 nm (excitation) and 510 nm (emission). Fluorescent intensities were normalized to total protein in cell lysate. EV release was measured using similar protocol with fluorescent intensities and total protein measured in the collected cell media.

### 4.6. EV Mass Spectrometry (LC/MSMS) and Proteomics Analysis

The proteomics profiling of EVs isolated from mIMCD and T2J cell lines cell culture media were outsourced to System Biosciences, Inc., Palo Alto, CA. Sample preparation and mass spectrometry were done performed as per manufacturer instruction. The ExoQuick ULTRA kit was used to isolate EVs from cell media. The kit uses size exclusion columns that show better purification of EVs compared to other methods [49]. Briefly, 10μg of each sample were processed by SDS-PAGE. Followed by gel digestion by nano LC-MS/MS with a Waters NanoAcquity HPLC system interfaced to a ThermoFisher Q Exactive mass spectrometer. Peptides were loaded on a trapping column and eluted over a 75 μm analytical column at 350 nL/min; both columns were packed with Luna C18 resin (Phenomenex). A 2hr gradient was employed. The mass spectrometer was operated in data-dependent mode, with the Orbitrap operating at 70,000 FWHM and 17,500 FWHM for MS and MS/MS respectively. The fifteen most abundant ions were selected for MS/MS. Data were searched using a local copy of Mascot (Matrix Science). Mascot DAT files were parsed into Scaffold (Proteome Software) for validation, filtering and to create a non-redundant list per sample. Data were filtered at 1% protein and peptide false detection rate and requiring at least two unique peptides per protein.

Quantitative proteomics data saved in Excel format were provided to the UTHSC Molecular Bioinformatics Core. Data were log2 transformed and normalized using a loess cyclic normalization limma package in R [50]. The mean, median, standard deviation, and coefficient of variance were calculated for every protein measured. Principle component analysis and Pearson’s coefficient plots were performed on the normalized proteome profile. A Wilcoxon’s t test was used to determine significant between conditions. All proteins that fail to yield a *p*-value less than 0.05 were removed. Benjamini–Hochberg false discovery rate was performed on the trimmed gene list [51]. All proteins that fail to yield a false discovery rate of less than 0.05 were removed. The final significant differential protein list was loaded into R to generate heatmaps. The targets were loaded into iPathway guide for pathway analysis, biological process identification and disease association [52]. We also compared the proteins identified in the mIMCD cell lines reported here with previously published EVs data, including EVpedia (http://student4.postech.ac.kr/evpe dia2_xe/xe/) and Vesiclepedia (http://www.microvesicles. org) databases.

### 4.7. Statistical Analysis

All experiments were conducted in triplicates. The data are presented as mean ± standard deviation. Student *t*-test was performed between the groups. The significance level was indicated as ****p* < 0.0005, ***p* < 0.005, and **p* < 0.05.

## Figures and Tables

**Figure 1 ijms-21-01729-f001:**
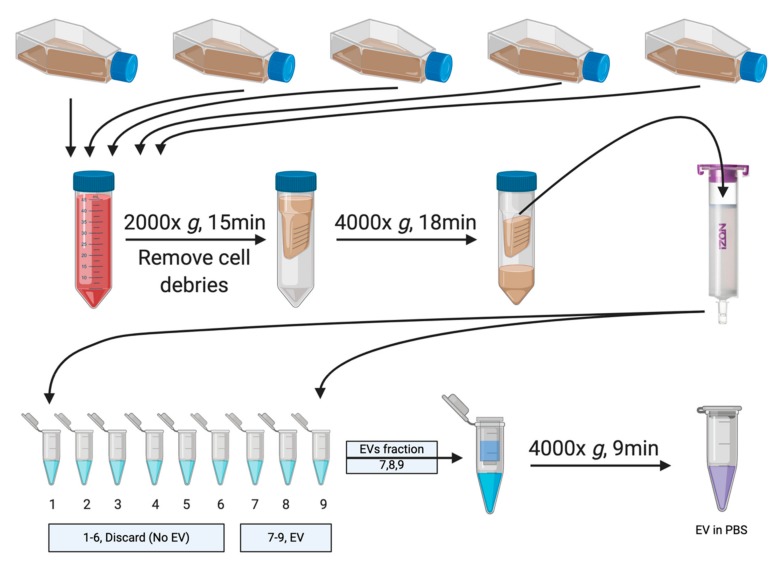
Schematic workflow of extracellular vesicle (EV) isolation. EVs were isolated from the serum-free media of inner medullary collecting duct (mIMCD) and T2J Cell lines (*Tsc2* deleted mIMCD cell lines). Pooled media were subjected to a low speed spin to remove the cells and cell debris. The cleared media were then transferred to a EV concentration device to bring the volume to 500 µL. This was then loaded onto the qEV size exclusion chromatography column. Elution was performed by gravity with fractions of 500 µL of sterile phosphate buffered saline (PBS) and the EV-rich fractions (seventh–ninth) were pooled and concentrated again using the concentrating device and used for downstream experiments.

**Figure 2 ijms-21-01729-f002:**
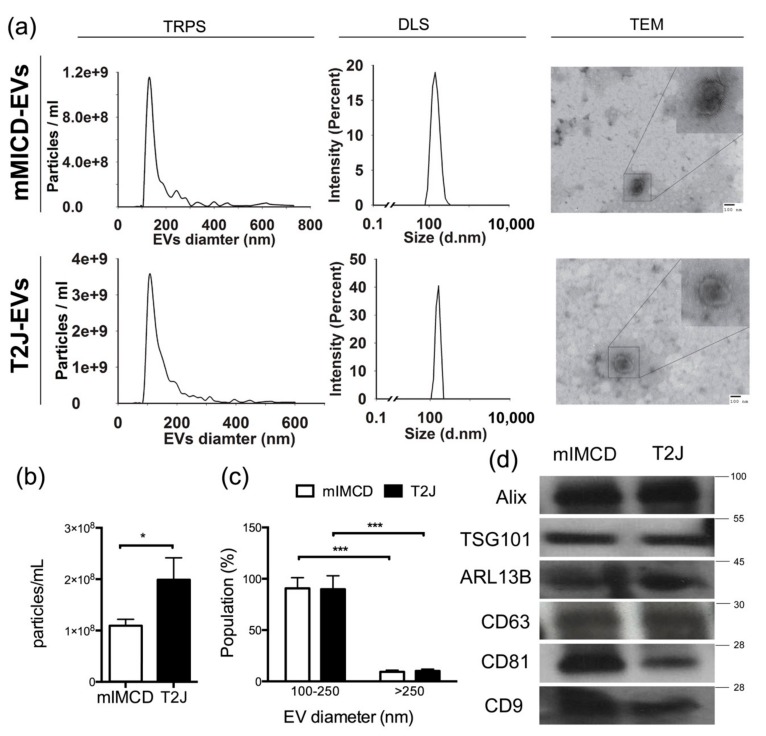
Characterization of mIMCD and T2J Cell-Derived EVs. (**a**) Three methods were used to describe size of EVs derived from mIMCD (top) and T2J cell lines (bottom). TRPS: tunable resistive pulse sensing, DLS: dynamic light scattering and TEM: transition electron microscopy. (**b**) Average concentration of EVs isolated form six independent cell media of mIMCD and T2J cells. (**c**) Approximately 88–91% of total EVs Recovered from mIMCD and T2J cell lines are smaller in size (100–250nm). (**d**) Six different final elutes from each cell lines were pooled together and concentrated, a maximum volume of 42 µL of EVs suspension were lysed and separated. Western blot analysis of EVs isolated from conditioned media of mIMCD and T2J cell lines shows the presence of traditional EVs markers Alix, TSG101, CD63, CD81, and CD9. Note: (* *p* < 0.05, and *** *p* < 0.001).

**Figure 3 ijms-21-01729-f003:**
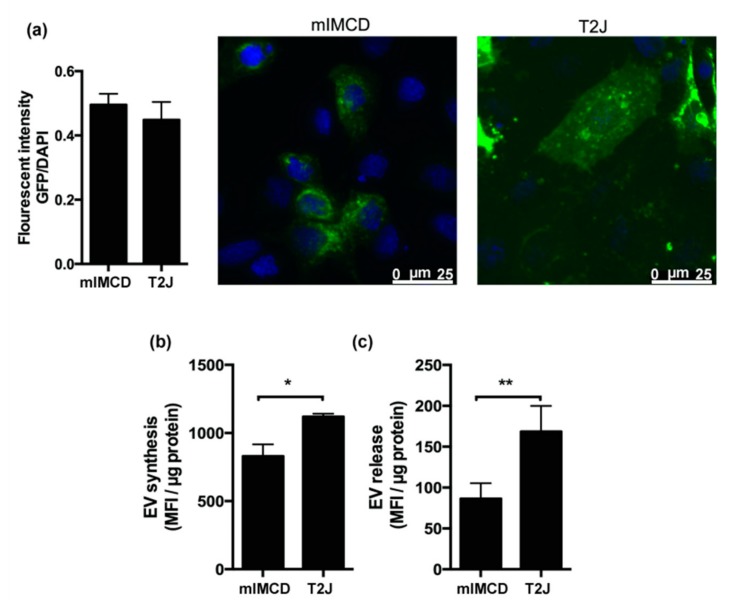
Extracellular vesicles synthesis, release and uptake. mIMCD and T2J cells were transfected with green fluorescent protein (GFP) tagged CD63, marker of extracellular vesicles (EVs). (**a**) Transfection efficiency was tested in 96-well culture plate by measuring mean fluorescent intensity of GFP normalized to intensity of nuclear 4′,6-diamidino-2-phenylindole (DAPI) stain (*n* = 3). Example of merged GFP and DAPI slide shown on right. (**b**) Synthesis of EVs from mIMCD and T2J cells measured 48 h after 0.2 μg GFP-CD63 transfection in a 12-well cell culture plate. After multiple washes, cells were lysed in lysis buffer and split for measurement of GFP fluorescent intensity and protein content (*n* = 3). (**c**) EV release mIMCD and T2J cells measured from debris-free cell media. GFP-fluorescence was normalized to protein content in cell media (*n* = 5).

**Figure 4 ijms-21-01729-f004:**
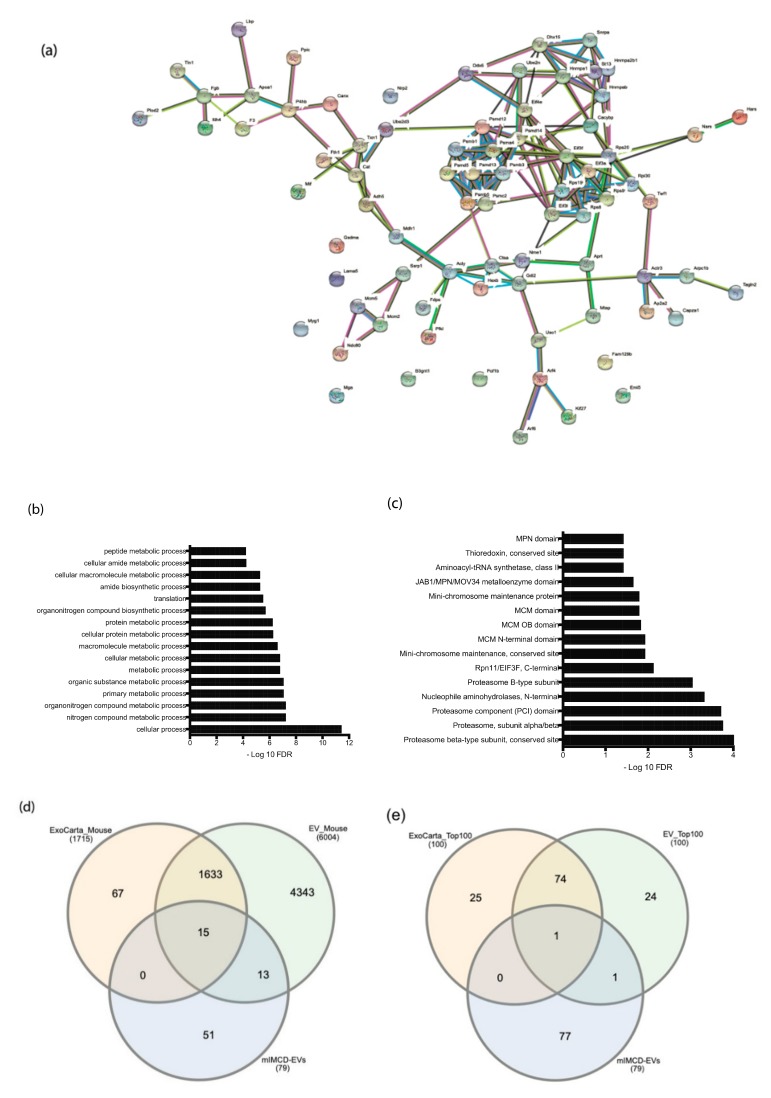
Analysis of mIMCD EV proteins. (**a**) Protein–protein interaction (PPI) network analysis of mIMCD EV proteins. There are significant clusters of interactions for proteolytic degradation, protein synthesis, and cell proliferation (see text). (**b**) Protein process enrichment, scale is −Log_10_ (False Discovery Rate). (**c**) Protein identity enrichment, scale is −Log_10_ (False Discovery Rate). (**d**) Comparison to available protein databases EVpedia and Vesiclepedia. (**e**) Comparison to the top 100 proteins in EVpedia and Vesiclepedia.

**Figure 5 ijms-21-01729-f005:**
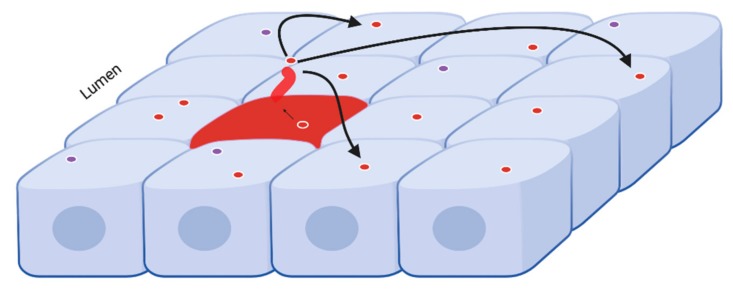
Model of EV signaling in renal tubule. Picture depicting the apical surface of tubule cells. The blue cells are unstressed intercalated cells, while the red cell depicts a principal cell (note primary cilia) experiencing physiological stress, such as caused by the loss of a *Tsc* gene activity. The darker red circles represent EVs from the stressed cell (red arrow), while the light blue circles represent EVs from cells that are not stressed. The EVs from the stressed cell carry messaging to help surrounding cells deal with the stressor, and to replace the cell that may succumb to the stress.

**Table 1 ijms-21-01729-t001:** Biological processes enriched by proteins in mIMCD EVs.

#Term ID	Term Description	Observed Gene Count	Background Gene Count	False Discovery Rate
	**Linked to Cell Proliferation**			
MMU-68827	CDT1 association with the CDC6:ORC:origin complex	8	53	6.19 × 10^–10^
MMU-69229	Ubiquitin-dependent degradation of Cyclin D1	8	46	6.19 × 10^–10^
MMU-69481	G2/M Checkpoints	10	128	6.19 × 10^–10^
MMU-69601	Ubiquitin Mediated Degradation of Phosphorylated Cdc25A	8	47	6.19 × 10^–10^
MMU-174154	APC/C:Cdc20 mediated degradation of Securin	8	61	9.92 × 10^–10^
MMU-69206	G1/S Transition	9	96	9.92 × 10^–10^
MMU-8948751	Regulation of PTEN stability and activity	8	64	1.25 × 10^–9^
MMU-174178	APC/C:Cdh1 mediated degradation of Cdc20 and other APC/C:Cdh1 targeted proteins in late mitosis/early G1	8	66	1.43 × 10^–9^
MMU-174184	Cdc20:Phospho-APC/C mediated degradation of Cyclin A	8	66	1.43 × 10^–9^
MMU-69017	CDK-mediated phosphorylation and removal of Cdc6	8	66	1.43 × 10^–9^
MMU-5687128	MAPK6/MAPK4 signaling	8	67	1.52 × 10^–9^
MMU-8852276	The role of GTSE1 in G2/M progression after G2 checkpoint	8	68	1.58 × 10^–9^
MMU-69620	Cell Cycle Checkpoints	11	240	6.48 × 10^–9^
MMU-69278	Cell Cycle, Mitotic	10	435	1.18 × 10^–5^
MMU-5663213	RHO GTPases Activate WASPs and WAVEs	2	32	0.0154
MMU-5674135	MAP2K and MAPK activation	2	36	0.0186
MMU-6806834	Signaling by MET	2	63	0.0465
	**Linked to Primary Cilia**			
MMU-5358346	Hedgehog ligand biogenesis	9	58	2.87 × 10^–10^
MMU-4086400	PCP/CE pathway	9	82	6.19 × 10^–10^
MMU-4641258	Degradation of DVL	8	52	6.19 × 10^–10^
MMU-5610780	Degradation of GLI1 by the proteasome	8	52	6.19 × 10^–10^
MMU-5610785	GLI3 is processed to GLI3R by the proteasome	8	54	6.19 × 10^–10^
MMU-4608870	Asymmetric localization of PCP proteins	8	57	7.09 × 10^–10^
MMU-5632684	Hedgehog ‘on’ state	8	105	2.79 × 10^–10^
	**Stress Response**			
MMU-1234176	Oxygen-dependent proline hydroxylation of Hypoxia-inducible Factor Alpha	9	60	2.87 × 10^–10^
MMU-349425	Autodegradation of the E3 ubiquitin ligase COP1	8	47	6.19 × 10^–10^
MMU-2262752	Cellular responses to stress	13	327	1.43 × 10^–9^
MMU-3299685	Detoxification of Reactive Oxygen Species	3	32	0.00072

Pathway analysis and biological processes are obtained from Reactome biological processes.

**Table 2 ijms-21-01729-t002:** Loss of Tsc2 Function is Associated with a Significant Decrease in Four Proteins.

ID	mIMCD Mean	T2J Mean	Difference	Fold Change	*p*-Value
Myosin-9	12.54 ± 1.74	5.63 ± 3.26	0.45	2.23	0.02
T-complex protein 1 subunit γ	4.42 ± 0.59	2.49 ± 1.00	0.56	1.77	0.03
Adseverin	5.27 ± 1.41	2.06 ± 0.84	0.39	2.56	0.04
Protein disulfide-isomerase A3	2.95 ± 0.28	1.81 ± 0.57	0.61	1.63	0.02

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
