# Peer review of "Tuberous Sclerosis Complex Axis Controls Renal Extracellular Vesicle Production and Protein Content"

_ijms, 2020, doi:10.3390/ijms21051729_

Round 1

Reviewer 1 Report

In this manuscript the authors use inner medullary collecting duct cells to model the effects of Tsc2 deletion in TSC renal disease. They purify extracellular vesicles from control and Tsc2 mutant cells and analyse the size and rate of production of the EVs. They then perform proteomic analysis of the EVs and bioinformatic analysis of the proteins identified from control and Tsc2 mutant cell EVs. This is an interesting manuscript that suggests potential new mechanisms which may potentially contribute to TSC renal disease. However, generation and characterisation of the Tsc2 mutant cell line used needs to be described in detail to support the findings. I have the following comments:

-The only reference cited for the Tsc2 mutant T2J cell line is in the Materials and Methods and is a review article that does not describe these cells. Details of the CRISPR/CAS-9 method used for generation of the T2J cell line needs to be described. The T2J cells also need to be characterised to show whether they carry a homozygous or heterozygous deletion of Tsc2 and to show loss of TSC2 protein expression and activation of mTOR signalling by analysing P-S6K and P-4E-BP expression.

-Statistical analyses should be used to support the following statement in the Results: “We were consistently able to isolate about three-fold more EVs from the Tsc2-deleted T2J cell line compared to the parental mIMCD cell line”.

Minor comments:

-Figure 1 is first mentioned in the Materials and Methods and so should be included as a supplemental figure.

-Figure 1 legend: “The eluted were eluted…?”

-Figure 2 should be separated into panels. Figure 2a is mentioned in the Results but Figure 2 does not have a panel a.

-Labelling of the axes in Figure 2 are too small and needs to be enlarged.

-In Figure 3a, how are control and Tsc2 mutant cells both compared between the two particles sizes in one statistical test?

-Table 1 should be described in the Results.

-Discussion third paragraph: Figure 4 should be Figure 2.

-Figure 5f should be Figure 5e.

-Figure 6 is not mentioned in the text.

-Labelling of the scale bars in Figures 2 and 4 is too small.

-References [1] and [7] appear to be the same and names are duplicated in References [1] and [6].

Author Response

The only reference cited for the Tsc2 mutant T2J cell line is in the Materials and Methods and is a review article that does not describe these cells. Details of the CRISPR/CAS-9 method used for generation of the T2J cell line needs to be described. The T2J cells also need to be characterised to show whether they carry a homozygous or heterozygous deletion of Tsc2 and to show loss of TSC2 protein expression and activation of mTOR signalling by analysing P-S6K and P-4E-BP expression.

Reply: The reference has been corrected. Further CRISPR/cas-9 method details are supplied along with the reference.  This reference, in figure 2, also demonstrates the loss is shown Tsc2 protein.

Statistical analyses should be used to support the following statement in the Results: “We were consistently able to isolate about three-fold more EVs from the Tsc2-deleted T2J cell line compared to the parental mIMCD cell line”.

Reply:  We repeated concentration measurements for 6 independent EV isolates, Figure 2b is introduced and the statement has been corrected to reflect this.

Minor comments:

-Figure 1 is first mentioned in the Materials and Methods and so should be included as a supplemental figure.

Reply: Figure 1 provides illustrative description of EVs isolation. We now introduce the procedure in result section and it now precedes figure 2.

-Figure 1 legend: “The eluted were eluted…?”

Reply: Thank you, this has been corrected.  

-Figure 2 should be separated into panels. Figure 2a is mentioned in the Results but Figure 2 does not have a panel a.

Reply: We have corrected the figure and agree that this is much better.

-Labelling of the axes in Figure 2 are too small and needs to be enlarged.

Reply: The Figure is changed to a better one and the labeling size is larger.

-In Figure 3a, how are control and Tsc2 mutant cells both compared between the two particles sizes in one statistical test?

Reply: they were statistically not significant. Following statement is added to the result : ‘’The difference in particle distribution between mIMCD and T2J cells was not statistically significant.’’

-Table 1 should be described in the Results.

Reply: Table 1 is now introduced in result section.

-Discussion third paragraph: Figure 4 should be Figure 2.

Reply: This has been corrected to Figure2b.

-Figure 5f should be Figure 5e.

Reply: Thank you, this has been corrected.

-Figure 6 is not mentioned in the text.

Reply: This has been corrected and figure 6 is now mentioned in first paragraph of the discussion.

-Labelling of the scale bars in Figures 2 and 4 is too small.

Reply : Labeling has been increased in size.

-References [1] and [7] appear to be the same and names are duplicated in References [1] and [6].

Reply: There have been corrected.

Reviewer 2 Report

The manuscript "Tuberous sclerosis complex axis controls extracellular vesicles production and protein content" by Zadjali et al. describes alterations in the production of EVs following the silencing of Tsc2 in an inner medullary collecting duct cell line. This work is a logical extension of their previous work, which suggested a role for EVs in the renal cystogenesis associated with TSC. Clear evidence is provided that EV production and release are indeed increased in IMCD cells lacking Tsc2. Subsequent proteomics analyses suggest cell-type specific exosomal proteins as well as differences between Tsc2 wild type and deficient cells. Before recommending this article for publication, I do have some concerns and suggestions. 

Major concern:

  1. The proteomics results are a major component of the data, thus I am concerned that a purity control is not included in the protein analyses of the EVs, as required by ISEV guidelines. Although it is stated that the EV prep is pure, with references to methods by other investigators, no evidence is provided to demonstrate the absence/depletion of contaminants. This verification is especially importance since the proteomics data indicate that the majority of proteins identified have not been previously identified in EVs.

Minor concerns:

  1. The title seems to me a bit broad relative to the content of the manuscript. Perhaps consider including the cell type or "renal".
  2. Introduction, last paragraph: Some background on EVs in TSC and/or kidney disease would be helpful here as well as the connection between your previous study and EVs.
  3. Figure 1 legend requires editing for grammatical errors  
  4. Figure 1 is not referenced in the text until the Discussion. Consider re-ordering your figures.
  5. Figure 2 axes can not be read. X-axis label is missing on TRPS graphs. Y-axes should be easily readable or put on the same scale so the differences are obvious. 
  6. Consider swapping Fig 3a with the TEM images in Figure 2 to improve cohesiveness.
  7. Figure 6: is referenced in the text as Figure "5" in the second to last paragraph of the Discussion.
  8. section 2.3, line 2: uptake should be "release"
  9. Figure 5A is not referenced in the text
  10. Table 1 is not referenced in the text until the Discussion (after Table 2)

Author Response

We have responded in a Word document because we include a figure.

Round 2

Reviewer 1 Report

I'm happy with the modifications. Just a couple  of minor corrections to be made:

-There are some minor corrections still to be made to the references (e.g. Refs 1, 4 and 44). 

-There are 2 versions of Figure 2 (I assume the first version is the correct one) and Figure 3 is duplicated.

Author Response

These issues have been addressed.  Thank you for your review.

Reviewer 2 Report

My concerns have been addressed in the revision and authors' comments. Only minor text revisions are required (removal of duplicate figures on pages 24, 26, 27). 

Author Response

I have made sure there are no duplicate figures.